# Vitamin D and Bone fragility in Individuals with Osteogenesis Imperfecta: A Scoping Review

**DOI:** 10.3390/ijms24119416

**Published:** 2023-05-28

**Authors:** Maria Gnoli, Evelise Brizola, Morena Tremosini, Alessia Di Cecco, Luca Sangiorgi

**Affiliations:** Department of Rare Skeletal Disorders, IRCCS Istituto Ortopedico Rizzoli, 40100 Bologna, Italy

**Keywords:** osteogenesis imperfecta, adults, children, vitamin D, measurement, supplementation

## Abstract

Vitamin D affects several body functions, and thus general health, due to its pleiotropic activity. It plays a key role in bone metabolism, and its deficiency impacts bone development, leading to bone fragility. In osteogenesis imperfecta (OI), a group of hereditary connective tissue disorders characterized by bone fragility, additional factors, such as vitamin D deficiency, can affect the expression of the phenotype and aggravate the disorder. The aim of this scoping review was to assess the incidence of vitamin D deficit in OI patients and the association between vitamin D status and supplementation in individuals affected by OI. We searched the PubMed Central and Embase databases and included studies published between January/2000 and October/2022 evaluating vitamin D measurement and status (normal, insufficiency, deficiency) and supplementation for OI. A total of 263 articles were identified, of which 45 were screened by title and abstract, and 10 were included after a full-text review. The review showed that low levels of vitamin D was a frequent finding in OI patients. Vitamin D supplementation was mainly indicated along with drug therapy and calcium intake. Even if widely used in clinical practice, vitamin D supplementation for OI individuals still needs a better characterization and harmonized frame for its use in the clinical setting, as well as further studies focusing on its effect on bone fragility.

## 1. Introduction

Vitamin D has a key role in different metabolic and development mechanisms, acting in particular in bone remodeling, skin differentiation, and immune system regulation [1]. Its role in calcium homeostasis and bone metabolism is well characterized; it is known that vitamin D deficiency affects different body mechanisms and systems [1,2,3,4,5,6,7,8], and recently its status has been evaluated in disorders other than osteoporosis or bone diseases, such as diabetes [9], sleep disorders [10], rheumatic diseases [11], and COVID-19 [12].

Vitamin D deficiency is considered a worldwide diffuse condition in the general population [13], being observed in 40% of Europeans [14], and with a prevalence of 24% in the USA and 37% in Canada [15,16,17]. Other epidemiological studies have revealed that Vitamin D deficiency is also diffuse in specific countries or geographic areas worldwide even with different prevalence [18,19,20,21,22,23,24,25,26,27,28].

A cutoff level of 50 nmol/L (or 20 ng/mL) to define vitamin D deficiency has been established in guidelines by different societies (Endocrine Society Task Force on Vitamin D; Institute of Medicine (IOM, Washington DC, USA)) [29], and when serum/plasma 25(OH)D concentration is below 75 nmol/L (or 30 ng/mL) is considered insufficient [29,30].

Vitamin D exists in two forms: vitamin D3 or cholecalciferol, which is synthesized in the skin after exposure to sunlight or ultraviolet light, and ergocalciferol or vitamin D2 which is obtained by irradiation of plants or plant materials, or foods [30]. Vitamin D is mainly acquired by sunlight (90%) on the skin (deep layers of the epidermis) from 7-dehydrocholesterol and absorbed by the small intestine. In liver microsomes, hydroxylation occurs in 25-hydroxyvitamin D3 (25(OH) vitamin D), and then a second hydroxylation occurs in the kidneys by 1α-hydroxylase (encoded by *CYP27B1*) to active vitamin D (1,25(OH)2 vitamin D), also known as calcitriol. The active metabolite 1,25(OH)2D enters the cell and binds to the vitamin D receptor, the classic effect of 1,25(OH)2D on active calcium transport in the intestinal cells. There is also vitamin D-independent calcium absorption through passive diffusion which depends on the calcium gradient and then on calcium intake [31,32,33].

Decreased serum calcium and phosphate levels stimulate hydroxylation, while increased levels reduce hydroxylation. Active vitamin D (1,25(OH)2 vitamin D) directly stimulates renal tubular calcium reabsorption and increases intestinal calcium and phosphate absorption. Furthermore, vitamin D stimulates osteoblasts to increase cytokine synthesis, osteoclastogenesis, and bone resorption [33].

Thus, vitamin D acts in the skeletal system regulating calcium absorption in the smallintestine and, with PTH, bone mineralization and calcium homeostasis, promoting a positive calcium and phosphate net balance [1,32]. Premature and dysmature birth, pigmented skin, low sunshine exposure, obesity, malabsorption, and advanced age are risk factors for vitamin D deficiency [32].

A severe deficiency of vitamin D causes rickets and osteomalacia. Rickets may be hereditary or acquired by inadequate intake of dietary vitamin D leading to bone fragility [34,35]. In rickets cases, inadequate mineralization of the osteoid bone matrix by calcium salts (osteomalacia) can be observed. Furthermore, low levels of vitamin D lead to high PTH levels causing high bone turnover, bone resorption, and osteoporosis. Both mechanisms increase fracture risk [34,35].

Several genetic diseases can be associated with bone fragility. In particular, osteogenesis imperfecta (OI), also called ‘‘brittle bone disease’’ indicates a group of hereditary connective tissue disorders characterized mainly by bone fragility and long bone deformities [36,37]. The disorder is genetically heterogeneous; however, *COL1A1* and *COL1A2* mutations are causative of about 85–90% of cases of OI, while several other genes account for a small percentage of cases [38]. The clinical expression is variable and the original OI classification by Sillence et al., (1979) [39] included four types of OI, reflecting the clinical severity of the disease as mild (OI type 1), lethal (OI type 2), severely deforming (OI type 3), and moderately deforming (OI type 4).

The other 10–15% of the cases are caused by pathogenic variants in genes related to the biosynthesis, post-translational modification, and/or folding of type I collagen; abnormalities in collagen chaperones were first described, but genes related to defects in formation and bone homeostasis, bone mineralization, osteoblast differentiation have also been recognized as causative of OI [37]. In recent years, genes acting in regulated intramembrane proteolysis in bone development have also been identified as OI causative, expanding the molecular mechanisms of bone fragility in OI [36,38].

A few genotype–phenotype correlations are well-reported, and it is known that the same mutation can lead to different clinical expressivity of the disease; nevertheless, qualitative abnormalities of collagen type I in general are related to more severe clinical expression [40].

The original Sillence classification divided OI into four distinct types according to the clinical and radiological characteristics, but not by causative genes [38]. Since then, the genes related to OI have increased in number and OMIM database entries for OI include 22 OI types (I-XXII) [35] The original classification has been updated considering the newly identified causative genes, combining the clinical, molecular, and radiological features in a new nomenclature (Table 1) [41].

Collagen type I is secreted by osteoblasts and is the most abundant constituent of the bone matrix. The mineral component of the bone is located within and between the collagen fibers. In the extracellular space, the osteoblasts produce alkaline phosphatase leading to the formation of mineral crystals in the gap regions between the collagen molecules [42].

In OI, pathogenic variants in *COL1A1* or *COL1A2* genes cause quantitative or qualitative abnormalities in collagen fibers. In fact, in collagen-type-I-related OI the collagen molecules are over-modified, the collagen fibers are thinner, and the bone matrix is hypermineralized, leading to bone fragility [43]. The bone in OI shows an abnormal architecture, with a lower trabecular number and connectivity, and lower trabecular thickness and volumetric bone mass. All these features contribute to bone fragility in OI [43].

The gold standard for the drug treatment of moderate to severe OI is bisphosphonates, antiresorptive drugs acting to inhibit bone resorption by osteoclasts [44], and in recent decades, several studies have focused on bisphosphonate treatment for children and adults with OI [44,45,46,47,48].

In light of the recent molecular knowledge, other treatment approaches have been evaluated [49,50,51], such as the use of denosumab (a monoclonal antibody directed against receptor activator of nuclear factor kappa B ligand (RANKL)) [52] or sclerostin antibody [53]. Furthermore, clinical trials are focusing on the use of antibodies acting on TGF-beta signaling [54] and cell or gene therapy [55].

Long before the first clinical classification of OI was established and the pathogenesis of the disorder was known, some authors had already considered including vitamin D supplementation in the management of the disease [3,56,57]. It is known that vitamin D supplementation has a beneficial effect on reducing the complications associated with vitamin D deficiency, including the low bone mineral density of the hip, spine, and arm bones [58]. The rationale for the use of vitamin D supplementation was that its deficiency had been directly linked to an increased fracture risk and severity in children [59]. Supplementation given to children classified as vitamin D deficient could have clinically useful benefits for peak bone mass [59,60,61,62].

As for children and adults in general [59,61,63], in OI patients, an adequate level of vitamin D is important for maintaining bone metabolism balance in a condition already with an increased risk of fracture. The optimal management of OI depends on the early and correct diagnosis of the disease and includes a multidisciplinary approach with pharmacological therapy, orthopedic follow-up, occupational therapy, physiotherapy, dietitians, and social workers [60,63,64]. Adequate vitamin D apport is part of the management of OI as in other conditions with bone fragility [65,66,67,68,69,70].

The aim of this scoping review was to evaluate the association between vitamin D status and supplementation in individuals affected by osteogenesis imperfecta.

## 2. Material and Methods

### 2.1. Study Design

We followed the PRISMA-ScR guidelines (10) and Joanna Briggs Institute Methods manual for scoping reviews (11) as a reference to develop this study.

The review included the 5 following steps: (a) definition of the research question; (b) identification of relevant studies; (c) selection of the studies; (d) data chart; (e) data extraction and summary of the results.

#### 2.1.1. Definition of the Research Question

The formulation of the search string followed the PICO system (12):-P: individuals affected by OI;-I: vitamin D status, measurement, and supplementation;-C: healthy subjects;-O: vitamin D status and supplementation.

#### 2.1.2. Search and Selection of the Studies

The literature revision was performed in the PubMed Central and Embase databases from January 2000 to October 2022. The eligibility criteria were: (1) studies evaluating the use of vitamin D measurement, status, and supplementation in individuals with a genetic or clinical diagnosis of OI; (2) including randomized and non-randomized studies, observational studies, case reports, and case series; (3) studies written in English or Italian; and (4) studies on pediatric and adult populations.

Meta-analyses, book chapters, short communications, letters to the editor, and conference abstracts were excluded.

The following search string was formulated:

Pubmed Central: (“vitamin D” [MeSH Terms] OR “vitamin D” [All Fields] OR “Vitamin D Deficiency”[Mesh] OR “Vitamin D/therapeutic use”[Mesh] OR “ergocalciferols” [MeSH Terms] OR “ergocalciferols” [All Fields] OR (“ergocalciferols “[MeSH Terms] OR” ergocalciferols “[All Fields] OR” ergocalciferol “[All Fields]) OR (“cholecalciferol “[MeSH Terms] OR” cholecalciferol “[All Fields] OR” cholecalciferols “[All Fields] OR” colecalciferol “[All Fields]) OR (“calcitriol “[MeSH Terms] OR” calcitriol “[All Fields] OR” calcitriols “[All Fields] OR Vitamin D supplementation OR Vitamin D 1,25 OH OR Vitamin D low levels OR Vitamin D insufficiency)) AND (“Osteogenesis Imperfecta “[MeSH Terms] OR” Osteogenesis Imperfecta “[All Fields]).

Embase: #1 AND (‘25 hydroxyvitamin d’/dd OR ‘calcitriol’/dd OR ‘colecalciferol’/dd OR ‘vitamin d’/dd) AND (2000:py OR 2001:py OR 2002:py OR 2003:py OR 2004:py OR 2005:py OR 2006:py OR 2007:py OR 2008:py OR 2009:py OR 2010:py OR 2011:py OR 2012:py OR 2013:py OR 2014:py OR 2015:py OR 2016:py OR 2017:py OR 2018:py OR 2019:py OR 2020:py OR 2021:py OR 2022:py) AND ‘osteogenesis imperfecta’/dm AND (‘case report’/de OR ‘clinical article’/de OR ‘clinical study’/de OR ‘clinical trial’/de OR ‘cohort analysis’/de OR ‘controlled clinical trial’/de OR ‘controlled study’/de OR ‘cross sectional study’/de OR ‘human’/de OR ‘major clinical study’/de OR ‘observational study’/de OR ‘prospective study’/de OR ‘randomized controlled trial’/de OR ‘retrospective study’/de) AND ‘article’/it.

Two independent reviewers performed the literature search and article selection, and the titles and abstracts of all the studies were reviewed to determine their eligibility. In case of disagreement on the suitability of the paper, a third author was consulted. Then, the full-text versions of the selected papers were extracted.

### 2.2. Data Extraction

The relevant data were extracted in a predefined form including (1) general paper information: first author and year of publication, the country where the study was conducted, the study design, aims, and duration; (2) the study population characteristics; and (3) vitamin D level, status, or measurement, and the main study results.

## 3. Results

The PRISMA flowchart is shown in Figure 1 and the PRISMA-ScR checklist is reported in Appendix A. A total of 263 potentially relevant studies were identified, and of those, 218 were excluded after the title and abstract screening. Of the 45 remaining studies, screened according to the eligibility criteria, only 10 were included. Among them, three were cross-sectional, one was a clinical trial, two were case-control studies, and four were retrospective studies. The study selection procedure is shown in the PRISMA flowchart (Figure 1). The data extracted from eligible articles and a detailed summary are presented in Table 2.

In the Iranian series [71], vitamin D deficiency was observed in more than 40% of cases, but the mean levels of vitamin D in OI patients (even if with criteria for deficiency) were higher than in healthy controls, which might be related to supplementary consumption in patients.

Vitamin D levels classified as insufficient (20–30 ng/mL or 50–75 nmol/L), moderately deficient (20–10 ng/mL or 50–25 nmol/L), or severely deficient (<10 ng/mL or 25 nmol/L) were reported by two studies in OI individuals, regardless of the degree of disease severity [73,75].

Lower levels of vitamin D were described in a series of OI adolescents where the average concentration was similar for all OI types; moreover, the most severely affected patients had the lowest vitamin D levels [78]. The percentage of vitamin D deficiency reported by Edouard et al. [79] in a study that included 315 OI patients was similar to the percentage found in children and adolescents with bone fragility (about 20%) by Bowden et al. [80].

Bowden and colleagues evaluated 85 children with osteopenia or osteoporosis, 24 of them with an OI diagnosis, observing that vitamin D deficiency was prevalent in this population, regardless of the specific diagnosis of bone disease [80].

Winzenberg and colleagues obtained similar results about the prevalence of vitamin D deficiency, comparing OI patients with a control group of healthy individuals [60].

In a series of 97 OI patients from Norway, normal vitamin D levels were observed and only 10% of the patients showed osteoporotic T scores [77].

A percentage as high as 80% for vitamin D deficiency was found in 52 Brazilian OI patients by Zambrano et al. [73]. This prevalence did not differ statistically by OI type; nevertheless, a deficient or insufficient level of vitamin D was observed in 100% of OI type III patients [73].

A positive association between the BMD z-score and serum vitamin D was reported in the largest study selected that involved 315 Canadian OI patients affected by diverse severity levels of the disease (OI type I, III, or IV): serum vitamin D (25OH vitamin D) levels were associated independently with the LS-aBMD Z score [79]. In another study, even if more than half of the OI children had low lumbar bone mass [64,79], there was no association between vitamin D level and BMD parameters [64]. The same observation was made in a retrospective study of 71 patients, where no relationship between vitamin D and the indicator of bone mass was described [79].

Fracture and height are the main clinical features and outcomes of OI. Chagas and colleagues evaluated the nutritional status of OI patients and found that body composition is a relevant risk factor for fractures [76].

Interestingly, a positive correlation between vitamin D levels and height was observed, independent of the OI type [75]. In the series described by Zambrano and colleagues, this correlation was found in children who received vitamin D supplementation [73].

Wekre and colleagues observed that in adult OI patients bone turnover tended to be increased and osteoporosis and lower vitamin D levels were more prevalent in OI type III than in other OI types [77]. Similar data were found by Zambrano et al. in children and adolescents, observing that, even if the prevalence did not depend on the OI type, deficient or insufficient levels of vitamin D were observed in all cases affected by OI type III [73].

## 4. Discussion

The impact of vitamin D on bone metabolism and calcium–phosphate homeostasis is well documented in the medical literature [1,32,33]. Cases of rickets due to vitamin D deficiency have been reported since the 17th century [81]. More recently, the role of vitamin D in multiple body systems and several diseases has been revised; nevertheless, only a few studies have evaluated vitamin D status and collected specific data related to the effect of vitamin D supplementation in individuals with OI. In addition, most of these studies could not confirm the correlation between vitamin D status and the increase in the clinical severity of the disease. Even if the number of studies on this topic is limited, most of them highlighted insufficient or deficient levels of vitamin D in OI patients as a frequent finding and observed few correlations between vitamin D status and other factors.

Insufficient or deficient vitamin D levels were reported in a variable percentage (from 20% to 80%) in different studies [73,78,79,80], confirming that vitamin D deficiency is a prevalent condition in OI, as in the general population. In particular, the same prevalence of vitamin D deficiency/insufficiency as in children or adolescent patients with bone fragility was observed [60,80].

Vitamin D insufficiency or deficiency was found in OI patients regardless of the degree of disease severity [73,75], even if all type III patients showed insufficient or deficient levels in a series [73] or the most severely affected individuals in an adolescent cohort had lower levels of vitamin D [78] in another study.

These findings about the prevalence of insufficient or deficient levels of vitamin D in OI are in accordance with data in the healthy population, and the fact that vitamin D deficiency is a diffuse health concern. [13,14,15,16,17]. Some authors have suggested that patients with more severe diseases also have restricted mobility and so less exposure to the sun. This hypothesis could also explain the fact that no effect of seasonality on vitamin D insufficiency was found in some studies [73,78,80]. The season of assessment was not related to the vitamin D concentration [73], although not all studies reported this information. Height is one of the main clinical features that varies with disease severity in OI. Unlike what has been found for the severity of the disease, a positive correlation between vitamin D levels and height, not depending on OI type, was observed. [75]. The same correlation was found in another series with patients who received vitamin D supplementation [73].

These described series included pediatric populations, while available data about vitamin D status in adult OI patients are insufficient. In a series of 97 OI patients from Norway, normal vitamin D levels were observed [77] and only 10% of the patients showed osteoporotic T scores.

Moreover, in an adult OI patient series, bone turnover tended to be increased and osteoporosis and lower vitamin D levels were more prevalent in OI type III than in other OI types [77].

Correlations between vitamin D and other health factors, such as bone mass index, parathormone (PTH) levels, or body composition, were not observed in any of the studies.

A positive association between the BMD z-score and serum vitamin D was reported in two of the selected studies [71,79]; however, this result was not confirmed in other articles [63,78].

A negative correlation between serum vitamin D and PTH levels has been observed in OI [73] as well as a positive correlation with alkaline phosphatase according to previous reports [75,76,80,82]. However, this association was not confirmed in all the studies, i.e., in the Brazilian study with 52 OI children, no correlation with bone markers was found [73]. This inverse relationship between vitamin D levels and PTH has been reported previously in healthy populations [1,81,82,83,84,85,86,87,88].

These findings corroborate the higher bone remodeling markers in metabolic disorders and response to low vitamin D levels, regardless of the OI type [83].

The change in PTH level occurs as a physiologic response to low levels of vitamin D and leads to high bone turnover, bone resorption, and osteoporosis [31,32]. The inverse correlation between vitamin D and PTH levels may be the expression of the effect of vitamin D deficiency on bone health, as an additional factor contributing to low bone mass and so to worsening of the disease [32,33]. Moreover, monitoring and supplementation of vitamin D should be advised for managing pediatric patients with osteopenia or osteoporosis, and also for the management of OI [80].

Overweight or body composition are also related to vitamin D levels: in particular in the study by Wilsfold and colleagues, overweight was another risk factor for lower levels of vitamin D in OI [75]. Some studies also revealed an association between vitamin D levels and fat/overweight in both children and adults not affected by OI [82,83,84,85,86,87,88,89,90,91].

Nutritional status in OI patients, and in particular body composition, is a relevant risk factor for fractures [76]. Increased body fat had a negative effect on bones, according to the inverse association between total body fat and bone mineral content in children previously described [91,92,93,94,95,96,97,98]. This correlation was also found in a small OI series: type III OI patients showed a decreased lean body mass (LBM) compared to controls and presented a higher percentage of body fat. In addition, compared with patients with type I OI, those with type III OI presented with lower body mass, height, length, and lean body mass (LBM), and higher BMI and number of fractures. This is in accordance with the fact that weight was proven to be related to lower bone mass in children [97]. Other studies will be needed to evaluate if the overweight and vitamin D levels correlation is independent of other factors (severity of the disease, physical activity, or sun exposure), in particular in OI patients. Indeed only a few of the studies collected detailed information about nutritional status or body composition in OI.

A number of studies on OI have highlighted optimizing lifestyle factors and nutrition (including calcium and vitamin D) along with physical exercise, as part of the management of the disease and also for bone health in general [64,65,66,67,68,69,70,99,100,101,102,103]. The estimated effects on BMD of vitamin D supplementation are probably relatively modest; moreover, appropriate levels of vitamin D can optimize the benefits of bisphosphonate treatment in adults [75].

No conclusive data are available about supplementation, nor about the optimal dose of supplementation, in the case of normal levels of vitamin D. Plante and colleagues compared two treatment groups (2000IU and 400IU). The supplementation with vitamin D at 2000IU increased serum 25OHD concentrations in children with OI more than supplementation with 400IU, but no significant differences in LS-aBMD z-score changes were detected and only about 20% of the cases had baseline vitamin D levels < 50 nmol/L [74].

The main limitations of the studies are that they were conducted in small series, showed different severity in clinical expression in few cases evaluated, and different molecular bases/mutations in collagen type I underlying the disease. Furthermore, the outcomes were different between the studies, and some data were subjective as they were patient self-reported, which can impact the reliability of the results. In most of the series, concomitant treatment or evaluation during clinical trials with drugs for the treatment of the disease represent confounding factors in evaluating vitamin D level effects per se in the disease. The majority of the studies had no molecular characterization of the cases, and only two studies reported *COL1A1* or *COL1A2* genetic testing, but not other genes correlated to OI.

## 5. Conclusions

This scoping review of the current evidence related to the incidence of vitamin D deficit in OI patients and the association between vitamin D status and supplementation in individuals affected by OI suggests that low levels of vitamin D (deficiency and insufficiency) is a frequent finding in OI individuals with different OI types. Only a limited number of studies focused on vitamin D status and on the benefit of its supplementation in the OI population; vitamin D supplementation was mainly indicated along with drug therapy and calcium intake. In light of the evidence, vitamin D may be considered in the follow-up and management of OI patients, as part of a multidisciplinary approach [64,65,66,67,68,69,70,100,101,102,103,104].

Nevertheless, it is important to consider that the evidence presented on vitamin D supplementation for OI does not change the risk of fracture, the primary outcome of OI.

Even if widely used in clinical practice, vitamin D supplementation for OI individuals still needs a better characterization and harmonized frame for its use in clinical practice, as well as further studies focusing on its effect on bone fragility and homeostasis.

## Figures and Tables

**Figure 1 ijms-24-09416-f001:**
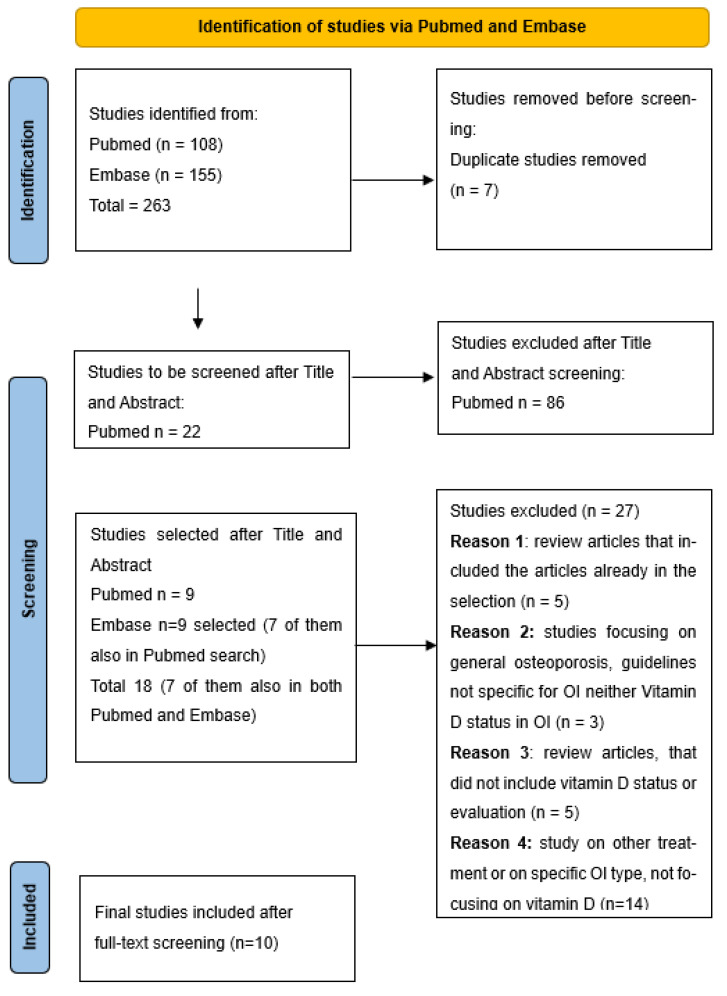
PRISMA flowchart.

**Table 1 ijms-24-09416-t001:** Update summary of OI correlated genes according to OMIM.

OMIM	OI Type in OMIM	Sillence OI Type	Inheritance	Defective Gene	Mechanism
166200	I	1	AD	*COL1A1, COL1A2*	Defects in collagen structure and processing
166210, 259420, 166220	II–IV	2–4	AD	*COL1A1, COL1A2*	Defects in collagen structure and processing
610967	V	3, 4 OI with calcification of interosseous membranes and/or hypertrophic callus (OI type 5),	AD	*IFITM5*	Bone mineralization defect
613982	VI	3	AR	*SERPINF1*	Bone mineralization defect
610682	VII	2, 3, 4	AR	*CRTAP*	Defect in collagen modification
610915	VIII	2, 3	AR	*LEPRE1*, (*P3H1*)	Defect in collagen modification
259440	IX	2, 3, 4	AR	*PPIB*	Defect in collagen modification
613848	X	3	AR	*SERPINH1*	Defect in collagen folding and cross-linking
610968	XI	3, 4	AR	*FKBP10*	Defect in collagen folding and cross-linking
613849	XII	4	AR	*SP7*	Osteoblast function and differentiation
614856	XIII	3	AR	*BMP1*	Procollagen processing
615066	XIV	3	AR	*T* *MEM38B*	Defects in collagen modification
615220	XV	3, 4	AD (*Osteoporosis*, WNT1-related)/AR	*WNT1*	Osteoblast function and differentiation
616229	XVI	3	AR	*CREB3L1*	Osteoblast function and differentiation
616507	XVII	3	AR	*SPARC*	Osteoblast function and differentiation
617952	XVIII	3	AR	*TENT5A*	Defect in BMP/TGFβ signaling pathway
301014	XIX	3	XR	*MBTPS2*	Osteoblast function and differentiation
618644	XX	3	AR	*MESD*	Defect in Wnt signaling
619131	XXI	3	AR	*KDELR2*	Defect in collagen folding and cross-linking
619795	XXII	3	AR	*CCD134*	Dysregulation of the RAS/MAPK signaling pathway

Abbreviations: AD, autosomal dominant; AR, autosomal recessive; XR, X-linked recessive. Note. Other forms (i.e., OI with craniosynostosis (Cole–Carpenter syndrome) or OI with congenital joint contractures, Bruck Syndrome) were not included; a complete table is available in the Nosology of genetic skeletal disorders: 2023 revision [41].

**Table 2 ijms-24-09416-t002:** Summary of included articles listed by year.

	Author Year Country Study Design	Population Characteristics	OI Type/Reported Severity/Genetic Testing	Vitamin D	Study Aim (s) and Main Results	Summary	Strength/ Limitation of the Study
1	Mohsenzade et al., 2021 [71] Iran Case control study	23 children affected by OI, 23 age–gender-matched controls; 9 males, 14 females	6 cases OI I 17 cases OI IV No molecular analysis data	Vitamin D deficiency was found in 43% of OI patients vs. 56% of controls Vitamin D levels were higher in OI patients (*p* = 0.033)	Aim: to assess the BMD and vitamin D level in children with OI in Iran Results: 43.4% of OI children had vitamin D deficiency No association between vitamin D levels and BMD parameters	Vitamin D deficiency is prevalent in OI patients.	Strength: -Case-control study assessing vitamin D status, BMD, and volumetric BMD in children Limitations: -Small sample -Self-reported data from a standard questionnaire -All the patients had received vitamin D supplements since the time of diagnosis
2	Nazim et al., 2019 [72] Egypt Case control study	26 children affected by OI, 26 controls; 13 males, 13 females in OI group	9 cases OI I 11 cases OI III 6 cases OI IV No molecular analysis data	25(OH) vitamin D lower than the reference range in 4 patients and > 100 µg/L in 5 cases. Note: In the study all patients were on vitamin D oral supplement	Aim: evaluate bone turnover markers in the Egyptian bone patients and the effect of bisphosphonate treatment in these markers Results: Serum calcium measurement, osteocalcin, P1NP are valuable for monitoring the effect of bisphosphonate treatment	4/26 patients showed low levels of vitamin D	Strengths: -Case-control study -Biochemical measurements, markers of bone formation, and markers of type I collagen degradation evaluation -Measurements at baseline, 6 months of treatment, and 12 months of treatment Limitations: -Small sample -Bisphosphonate treatment -Important variables not evaluated (number and location of fractures, Tanner stage, dietary vitamin D intake, and body composition)
3	Zambrano et al., 2016 [73] Brazil Cross-sectional study	52 patients affected by OI Age 1–19 y 29 females, 23 males	24 cases OI I 5 cases OI III 23 cases OI IV No molecular analysis data	Vitamin D deficiency was found in 35.5% and vitamin D insufficiency was found in 51.9% of OI patients; in 88.4% of cases vitamin D levels were insufficient or deficient	Aim: to assess the relationship between determinants of vitamin D status in pediatric patients with OI. Results: Vitamin D levels were insufficient or deficient in 88% of cases. Vitamin D levels were associated to LS- BMD z-score and were positive correlated to height. No significant difference in OI type No correlation with season of assessment No correlation with PTH or circulating bone markers was found	High prevalence of vitamin D low levels Correlation between vitamin D levels and LS BMD Z-score and height	Strengths: -Different outcomes assessed as vitamin D status, BMD, information about sun exposition, mobility, and bisphosphonate therapy Limitations: -Small sample -Blood samples collected in autumn/winter. -There are no longitudinal data -There are no data about vitamin D supplementation effects. -Important variables not reported (the number/location of fractures, Tanner stage, dietary vitamin D intake, and body composition)
4	Plante et al., 2016 [74] Canada Clinical randomized controlled trial.	60 individuals affected by OI Age 6 to 18.9 y; 35 females and 25 males Population was stratified for baseline bisphosphonate treatment and pubertal stage	23 cases OI 25 cases OI IV 12 cases OI III, V, or VI	Baseline vitamin D concentration, 80% > 50 nmol/L	Aim: to evaluate the efficacy of high-dose vitamin D supplementation on LS-aBMD in children with OI. Results: No significant differences in LS-aBMD z-score changes were detected between treatment groups Increase in vitamin D OH level after supplementation significantly higher in group receiving 2000 IU vitamin D	No significant differences in LS-aBMD z-score changes	Strengths: -Randomized controlled trial -Evaluation of vitamin D supplementation -Patients under bisphosphonate treatment in the previous 2 years were excluded Limitations: -No collected data reflecting endogenous vitamin D synthesis, such as skin pigmentation or sun exposure. -Simultaneous treatment with intravenous bisphosphonates in high proportion of participants
5	Wilsford et al., 2013 [75] USA Retrospective chart review	80 children with OI; charts of 44 children (26 female) had documentation of the variables of interest.	15 cases OI I 12 cases OI III 17 cases OI IV No molecular analysis data	Almost 80% of children with OI had insufficient or deficient levels of vitamin D	Aim: to evaluate the prevalence of vitamin D deficiency and possible risk factors influencing the vitamin D serum levels in patients with (OI). Results: Significant correlations with low vitamin D levels were found for older age (*p* < 0.001), African American descent (*p* = 0.01), BMI (*p* < 0.001), BMI percentile (*p* = 0.30), consumption of soda (*p* = 0.009), and pamidronate therapy (*p* = 0.004).	High prevalence of vitamin D deficiency or insufficient levels. Significant correlations with low vitamin D levels and BMI	Strengths: -Evaluation of several relevant parameters (season of year, level of ambulation, BMI, type of OI, time spent outdoors, and use of sunscreen before playing outdoors) Limitations: -Retrospective study -Missing number and location of fractures as main outcome. -Thirty-four (79.5%) patients had a history of pamidronate therapy
6	Chagas et al., 2012 [76] Brazil Cross-sectional study	26 patients affected by OI 13 type I OI and 13 type III OI 8 healthy controls Note: all patients were in treatment with pamidronate	13 cases OI I 13 cases OI III No molecular genetic testing information reported	69% type I patients 77% type III patients showed insufficient vitamin D levels 8% type III OI presented Vitamin D deficiency	Aim: Evaluate nutritional status, bone mineral density and biochemical parameters in OI subjects Results: in patients with OI number of fractures was positively related to body mass index and the percentage of body fat and negative correlated to lean body mass. Even taking dietary supplements, 12% of subjects did not achieve vitamin D recommendations	High prevalence of insufficient vitamin D levels in both type I and type III OI	Strengths: -Equal number of OI type 1 and type 3 patients. -First study in which a nutritional evaluation was performed in subjects with OI and body composition information collected. Limitations: -Small sample -Missing number and location of fractures as main outcome -No information about season -All patients were in treatment with pamidronate
7	Wekre et al., 2011 [77] Norway Case series	97 adult OI patients 41 males and 56 females Type I OI 74 Type III OI 9 Type IV OI 11 Unclassified 2	75 cases OI I 9 cases OI III 11 cases OI IV 2 unclassified cases No molecular analysis information	All patients showed normal levels of PTH, calcium and Vitamin D. OI type III displayed significantly lower values for 25 vitamin (OH) D (*p* = 0.05) than persons with type I and IV	Aim: Assess bone mass, bone turnover and prevalence of fractures in adult OI patients Results osteoporotic T scores in only 10% of patients Bone turnover markers were normal in the vast majority of patients. In adults with OI type III, bone turnover tended to be increased and osteoporosis more prevalent Seventeen persons (16 females and 1 male) were underbisphosphonates and/or hormone replacement therapy. There were no significant differences in anti-osteoporosis treatment between OI subtypes	Adults with OI type III, bone turnover tended to be increased, and osteoporosis more prevalent, and lower vitamin D levels than other OI types	Strengths: -Study in adult population -Prevalence and localization of fractures were evaluated Limitations: -Relatively small sample -Patient self-reported total number of fractures -No molecular analysis information -No information about bisphosphonate use in childhood -Other parameters (season, dietary vitamin D intake, sun exposure) not evaluated
8	Edouard et al. [78] 2011a Canada Retrospective study	71 patients affected with OI type I, III, or IV Age 1.4–17.5; 36 females, 35 males	29 cases OI I 12 cases OI III 30 cases OI IV In 63 patients a COL1A or COL2A3 mutation was identified (sequence analysis was performed in 65 patients)	Vitamin D deficiency in 52% of cases (Vitamin D concentration ≤ 50 nmol/L) Vitamin D concentration ≤ 80 nmol/L were found in 94% patients	Aim: to evaluated the relationship between vitamin D status and parameters of skeletal mineralization, mass, and metabolism in a group of pediatric osteogenesis imperfecta (OI) patients. Results: vitamin D was negative correlated with age and serum PTH levels No correlation with alkaline phosphatase levels. No seasonal variability Vitamin D levels were not related with bone formation rate, osteoid thickness, mineralization lag time. No evidence that vitamin D levels from 13 to 103 nmol/L were associated with measurement of bone mineralization, metabolism or mass in children with OI.	Deficient or low levels of vitamin D were found in more than 50% of patients Negative correlation between PTH levels and vitamin D levels was observed No seasonal variability	Strengths: -Histomorphometric parameters evaluated Limitations: -Small sample -Missing number and location of fractures as main outcome -No information about season or detailed information about treatment history
9	Edouard et al. [79] 2011b Canada Retrospective cross-sectional study	315 patients affected with OI type I, III, or IV Age 1.1–17.9 y; 161 females and 154 males	165 cases OI I 56 cases OI III 94 cases OI IV Collagen type I molecular testing available in 254 patients. Disease causing mutation in 222 patients	Vitamin D deficiency in 27% of cases Lowers levels in teenagers Levels decreased less markedly in winter than in other studies	Aim: evaluated vitamin D status determinants in children and adolescent OI patients Results: vitamin D levels were associated to LS-aBMD z-score in children and adolescents with OI, type I, III, IV. Vitamin D levels were inversely associated to PTH levels.	Vitamin D deficiency is prevalent in OI Lower levels of vitamin D were associated to LS-aBMD score and teenage.	Strengths: -Large sample size. -No previous treatment with bisphosphonate. Limitations: -No additional variables evaluated
10	Bowden et al., 2008 [80] USA Retrospective study	84 children with osteopenia or osteoporosis 24 OI patients (28% of the total)	There was no information about OI type or severity Some cases underwent collagen fibroblast analysis. No information about molecular genetics	Vitamin D deficiency was observed in 26% of OI cases Insufficient levels in 7% of OI patients	Aim: To determine the prevalence of vitamin D deficiency and insufficiency in children with osteopenia or osteoporosis and to evaluate the relationship between serum vitamin D levels and bone parameters, including bone mineral density. Results: A high prevalence of vitamin D insufficiency was found in this series of children with osteopenia or osteoporosis, regardless of the etiology of bone disorder. Negative correlation between vitamin D levels and PTH levels. No effect of seasonality on vitamin D.	High prevalence of insufficient or deficient vitamin D levels. Negative correlation between Vitamin D levels and PTH levels No effect of seasonality on vitamin D	Strengths: -Data about fracture rate concurrent with drug therapy -Demographic data, and -detailed medical history and biochemical laboratory studies Limitations: -Relatively small sample -Other disease with bone fragility included. -Other important variables not reported (i.e., type of OI or OI severity, location of fractures, season)

## Data Availability

Not applicable.

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
