# Peer review of "Vitamin D and Bone fragility in Individuals with Osteogenesis Imperfecta: A Scoping Review"

_ijms, 2023, doi:10.3390/ijms24119416_

Round 1

Reviewer 1 Report

This scoping review on the vitamin D status and bone fragility in patients afflicted with osteogenesis imperfecta (OI) has been performed according the rules. The few retained clinical studies highlight the vitamin D deficiency/insufficiency in these patients.

Introduction

The authors mention that 22 genes causing OI have been identified, leading to a new nomenclature. It would help to have a table summarizing this new nomenclature as well as giving the precise related genes and their mechanism of action.

Results section

The 10 selected studies are presented in a table form only without comments on their strengths & weaknesses. Furthermore, the table does not provide the type of OI that were considered. This is important as the severity of bone abnormalities is linked to the type of mutation.

Discussion

The discussion needs restructuration. Many of the comments provided could be placed in the results section. There are also some redundancies.

Minor comment

The mention that most severely affected patients (more severe form) have lower vitamin D levels (lines 272-274) stands to reason as these patients are less likely to have an adequate sunlight exposure.

The text would benefit from some syntactic editing.

Reviewer 2 Report

Based on the current status that vitamin D acts in skeletal system and a severe deficiency of vitamin D could cause rickets and osteomalacia. Meanwhile, Osteogenesis Imperfecta (OI) which also called ‘‘brittle bone disease’’ indicates a group of hereditary connective tissue disorders characterized mainly by bone fragility and long bone deformities. In OI patients an adequate level of vitamin D is important to maintain bone metabolism balance in a condition already with increased risk of fracture. Thus, the aim of this scoping review was to evaluate the association between vitamin D status and supplementation in individuals affected by Osteogenesis Imperfecta.

In general, this scoping review has revealed that low levels of vitamin D (deficiency and insufficiency) is a frequent finding in OI individuals with different OI types. However, only 10 articles were included after full-text review, whether this would lead to weak persuasiveness for this article. What’s more, the main limitations of the studies are that they were conducted in small series, showed different severity in clinical expression in few cases evaluated, different molecular bases/mutation in collagen type I underlying the disease, the reliability of the results is not guaranteed. Moreover, if the data could be presented in the form of tables or images, this article would be more persuasive.

There are some minor issues in this article that need attention and improvement. The title of “MATERIAL AND METHODS” is labeled with a serial number, but the “INTRODUCTION”, “RESULTS” and “DISCUSSION” are not labeled with serial numbers, the format of titles should use a unified format. In addition, there are some bold words or phrases in the “DISCUSSION” section, but they seem to have no special meaning, which makes it confusing.

Reviewer 3 Report

The reviewer enjoyed reading this manuscript and has a few comments:

1) This is a very good and extensive review on vitamin D deficiency with a special focus on its effect on bone and its relationship with cases of osteogenesis imperfecta. Because of the high prevelance of vitamin D deficiency all over the word, such an up-to-date review is really needed.

2) Since the review talks about supplementation of vitamin D, it will be interesting to add a few statements in the introduction or the discussion about the updated effects of over supplementation of vitamin D.

3) There is no need to insert the list of the 10 selected articles in page 6, line 231 & 232. These articles are well summarized in the tables in pages 7-11 and already included in the list of references (#64-73).

4) There are several formatting and English mistakes and the authors need to revise the text carefully. For example,

Line 10: there is an extra "may" that should be omitted.

Lines 18&19: the sentence needs to be rephrased.

Line 31: change the end of the sentense to: ...and immune system regulation.

Line 37: there is an extra "in"

... 

Lines 313 - 315: ..

Lines: 360 - 363: the sentence is toooo long!

There are several formatting and English mistakes and the authors need to revise the text carefully. For example,

Line 10: there is an extra "may" that should be omitted.

Lines 18&19: the sentence needs to be rephrased.

Line 31: change the end of the sentense to: ...and immune system regulation.

Line 37: there is an extra "in"

... 

Lines 313 - 315: ..

Lines: 360 - 363: the sentence is toooo long!

Round 2

Reviewer 1 Report

A minor correction is suggested. Line 133: “A vitamin D adequate apport is part” to “An adequate vitamin D intake (or supply) is part”

Overall the text is fine. May require final checking